# Analysis of UAV Thermal Soaring via Hawk-Inspired Swarm Interaction

**DOI:** 10.3390/biomimetics8010124

**Published:** 2023-03-17

**Authors:** Adam Pooley, Max Gao, Arushi Sharma, Sachi Barnaby, Yu Gu, Jason Gross

**Affiliations:** 1Department of Mechanical and Aerospace Engineering, Herbert Wertheim College of Engineering, University of Florida, Gainesville, FL 32611, USA; 2Engineering (Robotics), Ira A. Fulton Schools of Engineering, Arizona State University, Tempe, AZ 85287, USA; mgao18@asu.edu; 3Department of Mechanical and Aerospace Engineering, College of Engineering, The Ohio State University, Columbus, OH 43210, USA; sharma.1084@osu.edu; 4Department of Computer Science, School of Engineering, University of New Mexico, Albuquerque, NM 87131, USA; sbarnaby505@unm.edu; 5Department of Mechanical and Aerospace Engineering, Statler College of Engineering and Mineral Resources, West Virginia University, Morgantown, WV 26506, USA; yu.gu@mail.wvu.edu (Y.G.); jason.gross@mail.wvu.edu (J.G.)

**Keywords:** swarm intelligence, thermal updraft, boid

## Abstract

A swarm of unmanned aerial vehicles (UAVs) can be used for many applications, including disaster relief, search and rescue, and establishing communication networks, due to its mobility, scalability, and robustness to failure. However, a UAV swarm’s performance is typically limited by each agent’s stored energy. Recent works have considered the usage of thermals, or vertical updrafts of warm air, to address this issue. One challenge lies in a swarm of UAVs detecting and taking advantage of these thermals. Inspired by hawks, a swarm could take advantage of thermals better than individuals due to the swarm’s distributed sensing abilities. To determine which emergent behaviors increase survival time, simulation software was created to test the behavioral models of UAV gliders around thermals. For simplicity and robustness, agents operate with limited information about other agents. The UAVs’ motion was implemented as a Boids model, replicating the behavior of flocking birds through cohesion, separation, and alignment forces. Agents equipped with a modified behavioral model exhibit dynamic flocking behavior, including relative ascension-based cohesion and relative height-based separation and alignment. The simulation results show the agents flocking to thermals and improving swarm survival. These findings present a promising method to extend the flight time of autonomous UAV swarms.

## 1. Introduction

Unmanned aerial vehicle (UAV) swarms consist of a group of many autonomous or remote-controlled robots that each use local information to complete a task [1,2]. These swarms often contain either fixed-wing, single-rotor, multi-rotor, or hybrid UAVs and have been proposed for use in disaster relief, search and rescue, and surveillance due to their reliability and robustness to failure [2]. Many robot swarms derive inspiration from nature, where animals take advantage of their collective intelligence, i.e., ants forage in colonies, and wolves hunt in packs [3,4].

Within a swarm of UAVs, each agent’s operational performance is limited by its flight time [5,6,7,8]. In recent years, research has been conducted to extend the flight time of individual autonomous UAV gliders by taking advantage of naturally forming thermal updrafts while soaring [9,10]. Thermal updrafts are pockets of upward-moving warm air caused by uneven heating of the ground or sea by the sun [9]. The vertical airflow offers a source of free altitude gain for aircraft without engines, such as gliders [11]. For example, in 2005, the glider SoLong was able to fly for 48 consecutive hours using thermals and solar power [6]. The AutoSOAR project also offered an impressive demonstration in which a UAV was demonstrated to fly 7.8 h with 3 h of time, exploiting 132 thermals over the course of seven flight experiments [12,13]. These benefits are impressive for single-agent UAVs, and our work explores the additional benefits for swarms.

Part of the challenge of using thermals lies in detecting them since the location and magnitude of thermals are time and altitude dependent. Research into hawks has given insight into methods for identifying thermals. Though hawks are typically solitary birds, they form flocks to identify thermals faster [14]. Hawks often locate thermals by observing the ascent of other hawks and adjusting their position within the flock to take advantage of the strongest thermals [14]. When one hawk finds a thermal, other hawks will soon join and circle within the updraft [14]. As such, it has been proposed that a swarm of UAVs could take advantage of these natural occurrences better than an individual UAV due to the swarm’s distributed sensing ability [9]. With the ability to take advantage of thermals, a swarm of UAVs would be much better equipped to approach long-duration applications, such as migration or search and rescue.

In order to have a swarm properly take advantage of thermals, an algorithm needs to be developed that would enable individual agents to mimic the flocking behavior of hawks. In the interest of simplicity and utility with limited resources, this algorithm will be built under the assumption that agents cannot communicate with each other explicitly. It is assumed that agents, much like hawks, will only be able to see each other.

There is much research on UAV swarm models, though no existing model leverages swarm intelligence for the purpose of using thermals. Despite this, enough work has been completed to create a foundation upon which a model for thermal-using agents may be built. One important example is the Boids model, originally developed in 1987 by Craig W. Reynolds to produce a swarm with visually appealing flocking behaviors [15]. This bio-inspired model simplifies the flocking behaviors of birds into three rules: a cohesion rule for grouping, a separation rule for static collision avoidance, and an alignment rule for dynamic collision avoidance. Each rule has an associated gain determining its strength relative to other rules. Each agent assesses its local environment and alters its motion from the calculated results of its behavioral rules [15].

Early works considered the use and design consideration for parallel simulation agents to use Boid rules and explore flocking behavior in UAVs [16]. Some research has also considered the design of UAV swarm mission planning. For example, a centralized–distributed hybrid approach was determined to be effective for applications that require centralized task allocation [17]. More recently, however, the field has moved toward the use of decentralized agent-level interactions and agent-level intelligence as opposed to centralized coordination. This shift was to address to address the scalability issues associated with communication constraints/delays and reliability; one example is the use of UAV swarms in delivery applications [18]. Our work follows this approach and considers the impact of agent-level interactions without centralized coordination, specifically for taking advantage of the discovery and exploitation of thermal updrafts for soaring.

The Boids rule foundation has been used and modified by others seeking to make swarm models for various purposes. One example is a model created using a set of modified Boids rules with the intent to improve a swarm’s localization ability. In this model, the agents were found to exhibit clustering behaviors in response to homing rules being added to the Boids rules and gains being dynamic [19]. In another model made for the application of mapping, adding additional behaviors, including boundary, boundary avoidance, velocity gradients, and eccentricity, caused a great increase in the efficiency of mapping tasks [20]. Others have considered adding genetic algorithms for the application of increasing the durability of the communication networks established by a UAV swarm [21]. Following these recent developments from exploring the use of Boids and modified agent-level interaction rules, our work specifically considers an analysis of the Boids behaviors and variants for the specific application of exploiting thermals to extend the operational lifetime of the swarm through soaring.

This paper offers the contribution of providing insights as to which variants of the Boids rules applied to a swarm of UAVs result in better collective usage of thermals. In particular, the sensitivity of swarm survival with respect to several key design parameters is demonstrated by using a simulation environment with realistic assumptions. As an additional contribution of this work, the software to simulate the proposed swarm model is available for use by other researchers.

The rest of this paper is organized as follows. Section 2 explains the basis of the simulation: the problem statement, agent model, thermal updraft model, map design, simulation infrastructure, and agent behavior model. Section 3 discusses the results and trends from the simulation. Finally, concluding comments are provided in Section 4.

## 2. Methodology

### 2.1. Problem Statement

The problem considered is how to extend the average survival of swarm agents in a field containing unknown thermal energy sources. To observe UAV swarms intelligently using thermal updrafts, a simulation was created to facilitate the analysis of UAV swarms interacting with thermal updrafts. To enable the execution of these simulations, the following assumptions were made.

The simulated UAVs, or agents, are fixed-wing unpowered gliders modeled after the physical SB-XC sailplane [22,23]. The agents are equipped with a controller and stereo cameras. The agents can only “see” a specified number of agents within a specified radius. All agents are introduced at the start of the simulation and placed randomly.

Additionally, it is assumed that agents can observe local agents’ relative positions and velocities. Similar to the way in which hawks watch for the ascent of other hawks to locate thermals, it is considered essential for agents to know other agents’ relative positions and velocities to determine whether they are relatively ascending or descending.

To describe the data encapsulation and organization within the simulation, Figure 1 contains a flowchart of the information transfer between the batch simulation program, individual simulations, and individual agents.

While the simulation models and implementations will soon be discussed more deeply, a high-level description of the program would explain that the master script loads in all specified parameters and generates unique parameter combinations to be simulated by a main script. Each main script manages its own thermals and agents, which are both managed by the high-level classes ThermalMap and Swarm, respectively. Each agent calculates its local neighbors and its next state using its current state, the thermal strength at its position, and the states of its neighbors. When all agents die or the simulation reaches its maximum duration, the main script will return its simulation results to the master script to be consolidated and saved for future analyses.

### 2.2. Agent Model

The agent controller determines the behavior of each individual agent by using information from its local environment, such as which locally surrounding agents are ascending or where the agent is located on the map. An ascending nearby agent provides an indication of the existence of a thermal at that location. The local environment is defined using both the distance within which an agent can detect other agents as well as the maximum number of agents that can be detected. The lack of global information ensures that the agents require less computational power and are more applicable to information-deprived environments.

Local information influences the controller’s decision process and determines the agent’s next forward speed and bank angle. To simplify the model, aerodynamic approximations were applied from the literature [22], detailing that an agent’s vertical speed can be calculated as a function of the agent’s forward speed and bank angle. As a result, the forward speeds and bank angles close to their extremes are penalized with less lift.

### 2.3. Thermal Updraft Model

Thermal updrafts only appear during the day, and the timing and location of their formation can vary due to meteorological conditions, including the season and amount of cloud cover [24]. To reasonably simulate an environment that is applicable to the physical world, previous simulation work has simplified the thermal updraft model [25]. The thermal updrafts were modeled as cylindrical fields with vertical updrafts that add to an agent’s upward velocity depending on the agent’s position within the thermal. The modeled thermals, as with their real-world counterparts, vary in size and strength and are strongest in the center. Following a previous model in [25], a modified Gaussian curve was used to model the thermal updraft strength as a function of the distance from the thermal’s center. From the modified Gaussian curve, there is a ring around the center in which agents will experience an additional negative vertical velocity, which represents the region of downdrafts that encases the upward-moving air of a thermal.

In the simulation, it is assumed that thermals are spatially stationary and cannot overlap. This assumption stems from the lack of significant research on the dynamics of moving thermals and the results of thermals colliding. Furthermore, each thermal’s diameter and location are varied to ensure that the swarm’s emergent behavior is independent of the properties of individual thermals. Each thermal fades in and out based on randomized parameters, since real thermals fade in and out based on the time of day and land topography. In choosing the simulated dynamics of the thermals, the goal was to randomize the thermal parameters within a realistic range, estimated through research [26], resulting in thermals with varying sizes, strengths, locations, and fade cycles.

The experienced updraft speed of a thermal is calculated as a modified Gaussian curve. $v_{thermal}$ is the experienced updraft speed, $v_{max}$ is the maximum updraft speed at the thermal’s center, *d* is the distance from the center of the thermal, and *r* is the radius of the thermal. The calculation for a thermal’s updraft speed is expressed as:(1)$$v_{thermal}=v_{max}\left(1-{\left(\frac{3d}{r}\right)}^{2}\right)e^{-{\left(\frac{3d}{r}\right)}^{2}}$$

### 2.4. Map Parameters

The simulated area above which the agents fly is assumed to be flat ground, without weather, and has widths varying between 4 and 16 km, with a height range from 0 to 2600 m. First, flat ground was considered an appropriate assumption, simplifying the physical simulation of the thermals and ensuring that swarm emergent behavior is independent of the land topography. Second, the weather is clear to prevent agents from using cumulus cloud formation as a visual indicator of thermal updrafts [27]. This assumption sought to explore the efficacy of the agents’ control algorithms without environmental visual indicators for thermal updrafts. Third, the size of the simulated environment was chosen to encapsulate both the scale of thermal updrafts and the mobile range of an airborne glider.

### 2.5. Simulation Infrastructure

To explore the emergent behavior of the UAV swarm using thermal updrafts and to observe any dependencies between such behavior and the large domain of input parameters, many simulations were executed. The simulations were created in MATLAB and defined using 76 parameters. All parameter values were chosen after extensive background research, basing most values on real-world measurements and approximations. A complete listing of simulation parameters is included in Appendix A Table A1.

The simulations were programmed to read in all parameters from a configuration file at run time. To observe the effects on swarm emergent behavior, parameters were defined with a range of candidate values, and a batch of simulations was configured and run for each unique combination of all of the specified parameter values. When all 76 of the possible input parameters were discretized, the total number of simulations quickly exceeded any amount that could be tested within a reasonable amount of time. As such, only a small number of input parameters that were determined to be crucial were varied within a set of simulations. All other input parameters were held constant at specific values. The batch of simulations considered in this paper explored the effects of eight parameters, which are listed in Table 1. This batch of simulations sought to explore the effects of the number of agents, the number of thermals, the RNG seed, the cohesion gain, the separation gain, the alignment gain, the migration gain, and the cohesion distance exponent on the swarm’s survival and the simulations’ output metrics.

At the beginning of the simulation, the environment is generated, including the map of a specified size; a specified number of agents initialized to random positions and altitudes within boundaries; and a specified number of thermal updrafts initialized to random positions, sizes, and strengths within boundaries. Various map sizes have been tested, including 4 × 4 km, 8 × 8 km, and 16 × 16 km. The most implemented map size was 8 × 8 km due to its promising balance between map size and thermal size. When generating swarms, various swarm sizes have been explored, ranging from 5 to 80 agents. Swarm sizes of 20 agents and 40 agents were determined to produce the most interesting swarm interactions within a reasonable computational time. Simulations were conducted with 3, 6, 9, or 12 thermal updrafts due to the observed appropriate thermal densities in the environments and the rates at which swarms exploited the thermal updrafts.

Once the simulation starts running, the agent properties are updated. Agents use information about their local environment and their specified behavior model to calculate their desired velocity. Then, their desired velocity is separated into a forward speed and a bank angle. As described by the assumed aerodynamic model, the agent’s vertical velocity is calculated from its forward speed and bank angle. Finally, the updraft effects from the thermal updrafts are added to find the agent’s net vertical speed.

To simulate a decentralized swarm in which agents operate independently, each agent was represented as a unique instance of an agent class; each instance controlled its own properties. Each agent was individually updated by a higher-order swarm class instance and presented with only the local data that the agent would be able to observe, including the relative positions and relative velocities of other agents within its local environment.

### 2.6. Agent Behavior Model

Agents use a behavior model to determine their desired velocity, inspired by Boids rules. The original Boids model replicates the flocking behavior of birds with three rules: cohesion, separation, and alignment [15]. The agents use these three original rules and an additional migration rule. Each rule uses information from the agent’s local environment to calculate a desired velocity vector. The sum of each rule’s vector represents the agent’s desired velocity. A vector created from the cohesion rule will generally point toward the centroid of all local agents so that agents do not travel too far from the swarm. As most intelligent agent behavior is determined by nearby agents, a lone agent will not exhibit much intelligent behavior and is unable to contribute to the swarm’s goals. The separation vector will point away from local agents to avoid collisions. The alignment vector will direct an agent to match the velocities of nearby agents, dynamically avoiding collisions and resulting in the agents traveling in the same direction, mimicking the flocking behavior of animals in nature. Finally, the migration vector directs agents toward the center of the map. The migration rule was added to compel agents to stay within the map and near other agents, mimicking an animal’s natural ability to exhibit basic autonomy when separated from other animals. Agents separated from the swarm, without the migration rule, would lose all local stimulation and intelligent behavior, causing them to fly in a straight path, descending until they collide with the ground. The gains of all four rules are specified parameters that can be varied to change the agents’ emergent behavior.

Various methods of agent behavior models were tested, beginning with the original Boids model with migration. Modifications included height-based cohesion, separation and alignment, and testing the relative ascension-based cohesion. The final controller includes relative ascension-based cohesion, relative height-based separation and alignment, and migration. The details of this controller will be discussed in depth.

When an agent’s behavior rules calculate its desired velocity, it is influenced by other neighboring agents within its local environment. Other agents are considered to be within the agent’s local environment if the distance between the agents is less than a specified neighbor radius and if the other agent is within the k-nearest neighbors of this agent, where k is a specified parameter. Agents will consider two aspects of their neighboring agents: distance and velocity. A diagram detailing the relevant information between two interacting agents can be seen in Figure 2.

As previously described, an agent’s desired velocity, $\vec{F}$, is the sum of the output vectors from the agent’s behavior rules, including the cohesion vector, $\vec{C_{\mathrm{avg}}}$; the separation vector, $\vec{S_{\mathrm{avg}}}$; the alignment vector, $\vec{A_{\mathrm{avg}}}$; and the migration vector, $\vec{M}$. Each rule’s vector is weighted by a specified gain, including the cohesion gain, $C^{⋆}$; the separation gain, $S^{⋆}$; the alignment gain, $A^{⋆}$; and the migration gain, $M^{⋆}$, respectively. The calculation of the agent’s desired velocity can be expressed as:(2)$$\vec{F}=C^{⋆}\vec{C_{\mathrm{avg}}}+S^{⋆}\vec{S_{\mathrm{avg}}}+A^{⋆}\vec{A_{\mathrm{avg}}}+M^{⋆}\vec{M}.$$

As previously stated, the cohesion rule calculates an output vector, $\vec{C_{\mathrm{avg}}}$, which generally points toward the centroid of nearby agents. This output vector is calculated as the average of individual cohesion vectors to each local agent. For *k* nearby agents, a specific nearby agent n, and the individual cohesion vector to agent n, $\vec{C_{n}}$, the cohesion output vector can be expressed as:(3)$$\vec{C_{\mathrm{avg}}}=\frac{1}{k}\sum_{n=1}^{k}\vec{C_{n}}.$$

The individual cohesion vector to a specific nearby agent *n* is the product of three factors: height, relative ascension, and distance. The height factor serves to prioritize cohesion more at lower altitudes and less at higher altitudes. The relative ascension factor causes agents to coalesce more strongly to agents that are ascending relatively faster. The distance factor weighs cohesion to nearby agents more strongly than to farther agents.

The height factor is found using an agent’s altitude, *z*, normalized to the environment’s height ceiling, $z_{ceil}$, and raised to a height factor power, $P_{HF}$. The relative ascension factor takes the relative ascension of a neighbor, $\Delta v_{z}$, offsets it by the ascension ignore factor, $I_{AF}$, and normalizes it by the difference between the relative ascension maximum, $M_{AF}$, and the ascension ignore factor, $I_{AF}$. As with the height factor, the term is raised to the ascension factor power, $P_{AF}$. The relative position of the neighbor, $\vec{d_{\mathrm{Nbr}}}$, represents the difference in position between the neighbor and this agent. However, the distance factor uses the horizontal component of relative position, $\vec{d_{\mathrm{NbrXY}}}$, with magnitude, $|\vec{d_{\mathrm{NbrXY}}}|$, and direction, $\hat{d_{\mathrm{NbrXY}}}$. The distance factor is calculated as the magnitude of the horizontal relative position, $|\vec{d_{\mathrm{NbrXY}}}|$, scaled by the neighbor radius, $R_{Nbr}$, and raised to the cohesion power, $P_{C}$. These three factors are multiplied by the horizontal relative position’s direction, $\hat{d_{\mathrm{NbrXY}}}$, and used to calculate the individual cohesion vector, expressed as:(4)$$\vec{C_{n}}={\left(1-\frac{z}{z_{ceil}}\right)}^{P_{HF}}\times {\left(\frac{\Delta v_{z}-I_{AF}}{M_{AF}-I_{AF}}\right)}^{P_{AF}}\times {\left(\frac{|\vec{d_{\mathrm{NbrXY}}}|}{R_{Nbr}}\right)}^{P_{C}}\times \hat{d_{\mathrm{NbrXY}}}.$$

The separation rule calculates an output vector, $\vec{S_{\mathrm{avg}}}$, which generally points away from nearby agents. This output vector is calculated as the average of the individual separation vectors from each local agent. For *k* nearby agents, a specific nearby agent *n*, and the individual separation vector from agent *n*, $\vec{S_{n}}$, the separation output vector can be expressed as:(5)$$\vec{S_{\mathrm{avg}}}=\frac{1}{k}\sum_{n=1}^{k}\vec{S_{n}}.$$

Individual separation vectors are calculated using only distance but are masked by their relative altitude to the neighbor. This relative altitude masking addresses the fact that an agent’s motion is generally slower in the vertical direction, so potential collisions between agents at different altitudes are less concerning. If the magnitude of the neighbor’s relative vertical distance, |Δz|, is less than or equal to half of the separation gap, $S_{Gap}$, then the individual separation vector is calculated as given in Equation (Equation 6).
(6)$$\vec{S_{n}}=\left\{\begin{array}{cc} \langle 0,0\rangle & ,\;\;\left|\Delta z\right|>0.5\times S_{Gap}\\ -\hat{d_{\mathrm{NbrXY}}}\times {\left(\frac{|\vec{d_{\mathrm{Nbr}}}|}{R_{Nbr}}\right)}^{P_{S}} & ,\;\;\left|\Delta z\right|\leq 0.5\times S_{Gap}. \end{array}\right.$$
where $|\vec{d_{\mathrm{Nbr}}}|$; the magnitude of the relative position, $R_{Nbr}$, is the neighbor radius; $P_{S}$ is the separation power; and $-\hat{d_{\mathrm{NbrXY}}}$ is the negative horizontal relative position direction. The negative direction serves to direct the separation vector away from the neighbor. Note the use of the absolute relative position, rather than the horizontal relative position, to account for vertical separation. On the other hand, if the magnitude of the neighbor’s relative vertical distance is greater than half of the separation gap, the individual separation is set to zero, as indicated.

The alignment rule calculates an output vector, $\vec{A_{\mathrm{avg}}}$, which generally matches the velocities of nearby agents. This output vector is calculated as the average of the individual alignment vectors for each local agent. For *k* nearby agents, a specific nearby agent n, and the individual alignment vector for agent n, $\vec{A_{n}}$, the alignment output vector can be expressed as:(7)$$\vec{A_{\mathrm{avg}}}=\frac{1}{k}\sum_{n=1}^{k}\vec{A_{n}}.$$

Individual alignment vectors are also calculated using only the distance and masked by the relative altitude to the neighbor. If the magnitude of the neighbor’s relative vertical distance, |Δz|, is less than or equal to half of the alignment gap, $A_{Gap}$, then the individual alignment vector is calculated as the magnitude of the relative position, $|\vec{d_{\mathrm{Nbr}}}|$, scaled by the neighbor radius, $R_{Nbr}$, raised to the alignment power, $P_{A}$, and multiplied by the neighbor’s horizontal relative velocity, $\vec{v_{\Delta \mathrm{NbrXY}}}$. If the magnitude of the neighbor’s relative vertical distance is greater than half of the alignment gap, the individual alignment vector is masked out, expressed as:(8)$$\vec{A_{n}}=\left\{\begin{array}{cc} \langle 0,0\rangle & ,\;\;\left|\Delta z\right|>0.5\times A_{Gap}\\ \vec{v_{\Delta \mathrm{NbrXY}}}\times {\left(\frac{|d_{\mathrm{Nbr}}|}{R_{Nbr}}\right)}^{P_{A}} & ,\;\;\left|\Delta z\right|\leq 0.5\times A_{Gap}. \end{array}\right.$$

The migration rule calculates an output vector, $\vec{M}$, directing the agent to the center of the map. The output vector takes in a target position, nominally specified as the center of the map, and finds the offset between the target and this agent, $\vec{d_{\mathrm{Target}}}$, with magnitude, $\left|\vec{d_{\mathrm{Target}}}\right|$, and direction, $\hat{d_{\mathrm{Target}}}$. The output vector is calculated as the magnitude of the target offset, raised to the migration power, $P_{M}$, and multiplied by the direction of the target offset, expressed as:(9)$$\vec{M}={\left|\vec{d_{\mathrm{Target}}}\right|}^{P_{M}}\times \hat{d_{\mathrm{Target}}}.$$

The primary output metric used for swarm performance evaluation was the percentage of agents that survived the duration of the simulation, referred to as the survival percentage. The survival percentage is calculated using the number of surviving agents, *N*, and the initial number of agents, numAgents, expressed as:(10)$$SurvivingPercent=\frac{N}{numAgents}\times 100\%.$$

However, three other output metrics were measured to gauge various success characteristics of the swarm: the height score, the exploration percentage, and the thermal use score. The height score represents the success of the swarm to maintain high altitudes. The height score is calculated as the sum of all heights, *z*, of each agent, *n*, out of all *N* living agents, at every time-step, *t*, out of the maximum time-step, *T*, multiplied by the time-step duration, dt, expressed as:(11)$$HeightScore=dt\sum_{t=1}^{T}\sum_{n=1}^{N}z.$$

The exploration percentage quantifies the success of the swarm to explore the map. To calculate the exploration percentage, the simulated map is divided into segmented areas, and the percentage of segments the agents discovered is calculated as:(12)$$ExplorationPercent=\frac{DivisionsExplored}{DivisionsTotal}\times 100\%.$$

The thermal use score quantifies the success of the swarm to use thermal updrafts throughout the simulation’s duration. The thermal use score is calculated as the sum of all positive ascensions of each agent, n, out of all *N* living agents, at every time-step, *t*, out of the maximum time-step, *T*, multiplied by the time-step duration, dt. An agent’s positive ascension value is 1 if the agent’s vertical speed, $v_{z}$, is greater than 0 and 0 otherwise. The thermal use score is calculated as:(13)$$ThermalUseScore=dt\sum_{t=1}^{T}\sum_{n=1}^{N}\left\{\begin{array}{cc} 0, & v_{z}\leq 0\\ 1, & v_{z}>0 \end{array}\right\}$$

### 2.7. Simulation Hardware

The described methodology was carried out in MATLAB and executed on the WVU Robotics server cluster computer, consisting of two Intel Xeon E5-2698 v4 high-performance parallel compute nodes with 80 threads per node, running at a base speed of 2.20 GHz. As previously described in Section 2.5, the program was provided an extensive list of input parameters; each parameter either held constant or varied to explore that parameter’s effect on the swarm’s performance. The program generated unique combinations of the specified varying input parameters and simulated the swarm’s performance for each combination, appropriately distributing the simulations between all 160 threads of the cluster computer. As the program can automatically read the given input parameters and automatically save each simulation’s results, the batch simulation was carried out with very little user input once it began.

## 3. Results

The proposed modified Boids controller successfully enabled a swarm of autonomous UAVs to take advantage of thermal updrafts. The batch of 18,000 simulations with varying input parameters provided a large amount of insight into many properties of the observed swarm, including emergent behaviors, the ability to maintain the swarm’s altitude, and trends dependent on input parameters. To navigate the discussion of these results, this section will review the results of specific simulations and then consider the general properties and trends across all simulations.

### 3.1. Behavior Insights from an Individual Simulation Trial

To observe the effects of input parameters, as mentioned, a large batch of 18,000 simulations was run over an extended period of time. For simplified data analyses, each simulation recorded only a small amount of data, including the simulation’s given input parameters and its resulting output metrics. An example of a simulation’s recorded data, specifically chosen for this discussion, can be seen in Table 2. This simulation was chosen for discussion due to its high number of surviving agents and its final agent heights.

Despite the brevity of each simulation’s recorded data, individual simulations can be replayed using their input parameters to yield all requested simulation data. Because the input parameters completely define the simulation and all randomly generated elements, a simulation with identical input parameters will generate identical results. This facilitated both a high-level analysis of all simulations in the batch and a detailed low-level analysis of specific simulations replayed after the batch.

When analyzing the results of a simulation, visually observation of the swarm’s motion was found to be extremely helpful. An example video of a full-length simulation can be viewed here: https://youtube.com/playlist?list=PLKNuRrBV-8ciISteXaZpUXM1cUJ1svWiE. Replayed simulations can output all agent data at incremental time-steps, including position, velocity, and any requested controller data. Plotting the agents’ positions over time yielded insight into the emergent behaviors of the swarm. For the discussion of agent motion in a swarm with many survivors, the chosen simulation was replayed, and these plots were rendered with the resulting agent data, as seen in Figure 3.

As shown in Figure 3, the movement of all agents seems chaotic and unpredictable. However, by plotting the paths as partially transparent, distinct clumping appears in the form of five vertical columns at the locations of simulated thermal updrafts. These vertical clumps are the direct result of agents using thermal updrafts to gain altitude. Agents tend to move around the map less when attempting to use a thermal updraft for an extended period of time. This spatial stability creates an area of higher path density, visually identified as the vertical clumps, which stand apart from the lower path density areas created from a pseudo-random agent’s exploration.

To gain further insight into the behavior causing the paths rendered in Figure 3, the same paths were rendered again but colored using each agent’s vertical speed, as shown in Figure 4.

As shown in Figure 4, the plotted paths can be described as a uniform descent speed with focused areas of high ascent speeds, representing the thermal updrafts. As described by the thermal updraft model, areas of increased descent speeds can be observed around the thermal updrafts, correctly indicating the presence of downdrafts surrounding each thermal updraft. These plotted paths show that the agents can descend and successfully use thermal updrafts to ascend to a preferred higher altitude.

### 3.2. Discovery of a Thermal Updraft

The process by which a swarm can discover and use a thermal updraft to ascend to a higher altitude is a behavior of interest. As described by the agent UAV model, agents are unable to detect when they are in a thermal updraft. Instead, as described by the agent controller, agents coalesce more strongly toward agents with a greater relative ascension speed. As such, individual agents are unable to discover thermals, and, instead, groups of two or more agents must work together to discover thermal updrafts.

To gain insight into the behavior surrounding the discovery of a thermal updraft, two agents were selected from the chosen simulation due to the rapid speed at which they discovered and exploited a thermal updraft. Specifically, the positions of agents 19 and 30 were plotted, as shown in Figure 5.

As seen in Figure 5, agent 19 was flying semi-independently, agent 30 steered toward agent 19, agent 19 steered back toward agent 30, and finally, agent 30 steered back toward agent 19. To gain further insight into the behaviors causing these motions, these paths were plotted again but colored using the time, vertical ascension speed, and cohesion magnitude, as seen in Figure 6.

As shown in Figure 6, each time an agent actively steered, it steered toward the areas of the highest vertical ascension speed. Furthermore, each time an agent actively steered, the steering occurred at a local maximum of that agent’s cohesion magnitude, indicating that agent cohesion was primarily responsible for agents steering toward a thermal updraft. As agent 19 crossed over the thermal updraft, its vertical ascension speed increased, causing agent 30’s cohesion magnitude to increase and compelling agent 30 to steer toward agent 19. This process was found to repeat for both agents each time the other agent entered the thermal updraft. The process can be further observed by plotting the vertical position, vertical ascension speed, and cohesion magnitude of both agents against time, as shown in Figure 7.

Figure 7 shows that when one agent experiences a local maximum for vertical speed, the other agent experiences a local maximum for cohesion magnitude. Due to the oscillatory nature of the agent interactions, there is never a time when both agents are in the thermal updraft. When agent 19 exploits the thermal updraft, its high vertical speed compels agent 30 to coalesce toward the thermal updraft. However, before agent 30 can move into the thermal updraft, agent 19 begins exiting the thermal updraft due to agent 30’s low vertical speed. Then, agent 30 exploits the thermal updraft, compelling agent 19 to steer back toward the thermal updraft, thus repeating the cycle. The repetition of this process enables the successful exploitation of a thermal updraft.

Successfully discovering a thermal updraft requires favorable states of the first agents around the thermal, including close positions, velocities already steering close to the thermal, and a variety of favorable controller parameters that would cause the agents to coalesce to each other toward the thermal updraft. If the first agent to enter a thermal updraft is too far from surrounding agents, the first agent may completely leave the thermal updraft before any other agents have the chance to join. On the other hand, two agents could be so close to each other that they go through a thermal updraft simultaneously, causing them to experience the same vertical ascension and inducing little desire to coalesce toward each other. For the latter, a similar example can be seen in Figure 8, in which two agents completely pass through a thermal, and two nearby agents are left unable to use the thermal.

Figure 8 shows two paths cleanly passing through a thermal updraft and two paths steering near the thermal updraft but ultimately unable to use the thermal. To gain further insight into the behavior causing these motions, Figure 9 shows vertical position, vertical speed, and cohesion magnitude plotted against time for agents 9, 11, 27, and 29.

As shown in Figure 9, agents 9 and 29 passed through the thermal updraft, with agent 9 going first and agent 29 following shortly behind. As such, they both experienced higher cohesion magnitudes toward each other. However, due to the unfavorable conditions in which they passed through the thermal, both agents continued moving out of the thermal updraft. Agents 11 and 27 both experienced higher cohesion magnitudes toward the agents in the thermal updraft but were unable to reach the thermal quickly enough. With no agents in the thermal updraft to attract other agents, both agent 11 and agent 27 steered to avoid collision and left the area.

### 3.3. Exploitation of a Thermal Updraft

Another significant behavior of interest is the process by which large numbers of agents are able to exploit a previously discovered thermal updraft. Following the discovery of a thermal updraft, exploitation is considered to be the process by which new agents are drawn in to use the thermal and the agents already using the thermal continue to use it.

As previously discussed, the agent model and interaction controller generally dictate that agents actively move toward thermals only when at least one other agent is using the thermal and has the higher vertical speeds needed to attract nearby agents. The successful discovery of a thermal updraft guarantees this condition is met by the agents that discovered it. As more agents move near the thermal, they will naturally coalesce with the other agents within the thermal, as shown in Figure 10. This process is self-sustaining and will generally continue to repeat until the thermal fades away or until all agents become so high that they stop prioritizing altitude gain and stop coalescing with the agents within the thermal.

Individual agent motion is very chaotic and changes rapidly based on their environment. As such, an agent exploiting a thermal can still move toward the thermal’s edges, the thermal’s surrounding downdraft, or even leave the thermal completely. In terms of continuing the swarm’s exploitation of the thermal updraft, these positions do not attract agents into the thermal as well. If too many agents using a thermal updraft moved to these unfavorable positions, it would be possible to lose the thermal updraft altogether, similar to a failed discovery of a thermal. As such, a swarm’s exploitation of a thermal updraft greatly improves as the number of agents currently attempting to exploit the thermal updraft increases. When more agents exploit a thermal updraft, the exploitation of that thermal becomes more robust to random unfavorable movements of individual agents.

### 3.4. General Trends from Large Batch Simulation

To assess the controller’s success at allowing a swarm to take advantage of thermal updrafts, the percentage of surviving agents was calculated for each simulation trial. The percentage of surviving agents is more useful than the number of surviving agents because the total number of agents in each simulation was a varied input parameter. The results were collected into a histogram, shown in Figure 11.

As shown in Figure 11, a notable number of simulations resulted in a wide range of possible amounts of swarm survival, meaning that the tested input parameters contain combinations that can result in all agents surviving or no agents surviving. Specifically, 31 simulations resulted in all agents surviving, 4880 simulations resulted in at least 50% of agents surviving, and 7505 simulations resulted in all agents dying. Considering the survival effects from any individual input parameter, the massive spread in survival data is due to the combined efforts of all input parameters used in the simulation.

As previously discussed, almost 50% of all simulations in this batch resulted in no agents surviving, meaning that agent survival is extremely sensitive to the input parameters, and 0% agent survival is more likely than 100% agent survival.

#### 3.4.1. Sensitivity to Number of Agents

To assess the effects of the initial number of agents on swarm survival, histograms were created for the survival data generated with each value of the initial number of agents. These histograms were plotted based on the percentage of surviving agents, as shown in Figure 12.

As shown in Figure 12, larger percentages of agents survived when the simulations started with 40 agents than for the simulations with 20 agents. This result was expected, as many necessary swarm functions become more robust with an increasing number of agents. The significance of the larger percentages of surviving agents is that increasing the number of agents in a swarm increases the survival chances of all individual agents.

#### 3.4.2. Sensitivity to Number of Thermals

Similarly, to assess the effects of the number of thermal updrafts on swarm survival, histograms were created for the survival data generated with each value of the number of thermal updrafts. These histograms were plotted based on the percentage of surviving agents, as shown in Figure 13.

As shown in Figure 13, the percentage of surviving agents increases with the number of thermal updrafts. As previously discussed, agent discovery of a thermal updraft is very sensitive to the conditions of surrounding agents, and the success of any individual discovery is not guaranteed. However, if there are more thermal updrafts for agents to attempt to discover, successful discoveries will happen more often, leading to higher percentages of surviving agents.

#### 3.4.3. Sensitivity to Migration

To assess the effects of the migration gain on swarm survival, exploration, and thermal use, histograms were created for the output data generated with each value of migration. These histograms were plotted based on the percentage of surviving agents, the exploration percentage, and the thermal use score, as shown in Figure 14.

As shown in Figure 14, clear trends exist between the value of migration and the percentage of surviving agents, exploration percentage, and thermal use score. As the value of migration increased, the percentage of surviving agents consistently increased, the exploration percentage decreased, and the thermal use score increased. These clear trends indicate that the simulated swarm model’s performance has a high dependence on migration. When simulated with the lowest migration value, $1\times 10^{-23}$, agents had little incentive to stay within the map’s boundaries and consistently reached 100% exploration. As thermal updrafts are created within the map’s boundaries, the lowest migration value induced less use of thermal updrafts, which directly led to lower swarm survival. In contrast, the highest value of migration, $1\times 10^{-21}$, compelled agents to stay within the map’s boundaries, not exceeding explorations of 80%. The swarm’s compulsion to stay closer to the center of the map led to more thermal updraft use, which directly led to far greater survival percentages. Interestingly, although the exploration percentage was intended to help quantify a swarm’s success at exploring the map, this is a case where higher exploration percentages correlated to worse swarm performance, due to the lower survival percentages.

#### 3.4.4. Sensitivity to Cohesion Power and Alignment

Swarm survival was not significantly affected by cohesion power. In fact, swarms simulated with a cohesion power of 0 performed equally to those simulated with a cohesion power of 0.5. Cohesion power represents the exponent of the distance component of cohesion magnitude. A cohesion power of 0 would induce a cohesion magnitude that is independent of the distance to nearby agents. This indicates that the simulated model has room for simplification without a loss of swarm survival, as agent cohesion performs equally well without the ability to assess the distance to nearby agents.

Similarly, swarm survival was not significantly affected by alignment. Swarms were tested with alignment values of −0.001, 0, and 0.001, with equal swarm survival percentages from all values of alignment. Again, this indicates that the model can be simplified without a loss to swarm survival by removing the alignment rule from the agent model.

#### 3.4.5. Sensitivity to Cohesion and Separation

Due to the nature of the agent controller, cohesion and separation are often opposing forces. As such, it was understood that the effects of cohesion and separation on swarm survival should be assessed together. For each pair of cohesion and separation values, the mean percentage of surviving agents was calculated and plotted against both the cohesion and separation, as shown in Figure 15.

As shown in Figure 15, swarm survival trends for cohesion and separation are strongly related. Lower swarm survival percentages are yielded by poor combinations of cohesion and separation, characterized by large values of one variable and small values of the other variable. For the proposed swarm model, higher swarm survival percentages can be attained when the cohesion values are roughly 100,000 times greater than the chosen separation values. Optimal values for cohesion and separation lie toward the middle of their logarithmic ranges, with a peak mean swarm survival percentage of 34.73% generated with a cohesion value of 3000 and a separation value of 0.03.

This relationship between cohesion, separation, and their effects on swarm survival is relevant when trying to choose the cohesion and separation gains of the agent controller. Figure 15 shows that multiple pairs of cohesion and separation values perform comparably to each other, despite the large range in magnitudes. This indicates that when tuning the agent controller to improve swarm survival, the specific values of cohesion and separation matter less than the relationship between the chosen values.

### 3.5. Further Analysis of Metrics

To further observe the relationships between the input variables and the output metrics, the batch data from 18,000 simulations were plotted for thermal use, exploration percentage, and swarm survival, and colored for each unique value of the variable in question. These visuals enable an analysis of each input variable’s effect on multiple output metrics at once. These plots were generated and sorted by their observable variable effects in decreasing order of significance. Variables with large effects on the output metrics are shown in Figure 16.

As shown, the batch simulation results were heavily influenced by the migration gain and the initial number of agents. This correlates with the previous discussion of these metrics’ effects on each metric individually. Notably, the migration gain has a significant influence on the exploration percentage, creating large gaps in the output metrics along this axis, which are much larger than the effects of any other input parameter. The general observable trends are that as migration decreases, the swarm’s thermal use decreases, the exploration percentage increases drastically, and the survival percentage decreases. As such, the migration parameter provides an important utility to an operator controlling the behavior of the swarm, providing the ability to tune the swarm’s exploration of its surroundings.

The number of agents also had a significant effect on the output metrics, which is primarily apparent on the thermal usage axis. As the initial number of agents increased from 20 to 40, the swarm’s thermal use approximately doubled, the exploration percentage locally increased, and the survival percentage remained uninfluenced.

Variables with less significant effects, creating gradients within the results, are shown in Figure 17.

As shown, the batch simulation results were influenced by the cohesion gain, the separation gain, and the number of thermals. The influence of these variables created gradients within the results, consistent across different combinations of the values of migration and the initial number of agents. Although their effect is less significant than migration and the initial number of agents, these gradients do provide insight into the effects of cohesion, separation, and the number of thermals on the swarm’s performance.

The gradient appearing from cohesion expresses that as cohesion increases, the swarm’s thermal use also increases, with minimal impact on the swarm’s exploration or survival. The gradient appearing from separation expresses that as separation increases, the swarm’s exploration also increases, with minimal impact on the swarm’s thermal use or survival. The gradient appearing from the number of thermals expresses that as the number of thermals increases, the swarm’s thermal use and survival increases, with minimal impact on the exploration. The utility of these trends provides minor control of the swarm’s performance in these metrics, within a given combination of migration and the initial number of agents.

Variables with no discernible trends are shown in Figure 18.

These variables were found to have almost no effect on the simulation’s output metrics. This result is repeated for the alignment and cohesion power, which were individually found to have very little effect on the output metrics. Additionally, the lack of observable trends for the RNG seed is a promising verification of the swarm performing similarly under a variety of random initial conditions.

## 4. Conclusions

The simulations showed that modifying Boids-inspired behavioral rules to take advantage of thermal updrafts increases swarm survival. These modifications include relative ascension-based cohesion and relative height-based separation and alignment. Individual agents were unable to discover or exploit thermal updrafts, instead flying straight through them. However, groups of agents exhibited emergent behavior, making use of thermal updrafts by coalescing to ascending agents. Height, vertical velocity, and cohesion magnitude data lent insight into the sensitivity of discovering a thermal updraft, often preventing groups of agents from using a thermal updraft to regain altitude. Finally, trends between the input parameters and output metrics revealed characteristics of a swarm that lead to higher agent survival rates, including larger swarm sizes, more thermal updrafts, and stronger migration.

This work provides an analysis of a behavioral model that can be applied to UAV gliders in the real world. Agents have successfully exhibited a level of swarm intelligence that has bypassed any need for communication and centralized control, and, thus, the constraints and issues associated with such. Therefore, this model is compatible with a multi-UAV system not using any direct communication structure and may be implemented regardless of the type of communication hardware installed. Additionally, the agents were shown not to be hindered by a large population but rather to be hindered by the absence of other agents. Within the tested input parameters, the swarm bypassed issues associated with scalability, as the swarm’s effectiveness increased with size. Finally, these results contribute to the study of hawk behavior, which inspired the work of this paper. As the simulated UAV agents mimicked the behavior of hawks, the swarm’s emergent behaviors provide insight into the properties and benefits of hawk interaction.

### Limitations and Future Work

Due to limited computing power, many assumptions were made to enable the execution of the discussed simulations. Further research into the interactions between autonomous UAV swarms and thermal updrafts may seek to relax these assumptions and explore more unanswered questions.

The simulations were administered under limited environmental variation. Thermals were simulated close to the center of the map, and map boundaries were not enforced. This gave considerable significance to the agent migration behavioral rule, which kept the agents in close proximity to the center of the map and the simulated thermals. To prevent a strong agent dependency on migration, thermal updrafts could be simulated within a larger area, allowing the swarm to disperse considerably.

The thermal updrafts were modeled under ideal conditions as stationary, vertical cylinders with little altitude variation. A realistic model of the generation and propagation of thermal updrafts would strongly depend on the Earth’s topology and is outside the scope of this work. Applying a realistic thermal model would include the freedom for thermals to be more time-varying, altitude-varying, and subject to wind disturbances, allowing the thermals to drift as they rise. Exploring the performance and behaviors of an autonomous UAV swarm with a more realistic thermal model would likely induce new emergent swarm patterns, more closely mimicking real hawk behaviors, and further bridge the gap between simulation and hardware experimentation.

From the results of the simulations, many trends have been observed, including the effects of the swarm’s size, the number of thermal updrafts, and the migration gain. However, these trends may lose accuracy with a larger sample of variable values. For instance, while swarms with 40 agents had better survival rates than swarms with 20 agents, there may be an upper limit to the number of agents that can be supported by a given swarm or by a given environment. Further simulations may seek to explore the effects of input parameters with a higher range and resolution.

Finally, all simulations were explored under a large set of fixed parameters, as listed in the Appendix A, in Table A1. Although a considerable effort was put into choosing the fixed parameters that characterize the simulations, the discussed trends and emergent behavior may fail to generalize to swarms with different chosen parameters. Simulations executed with more varied parameters would serve to explore how these results generalize to other swarms, including swarms operated in the field, where aircraft models other than the SB-XC are used. Such work would require extensive computing power but would improve the application of this work to real swarms.

## Figures and Tables

**Figure 1 biomimetics-08-00124-f001:**
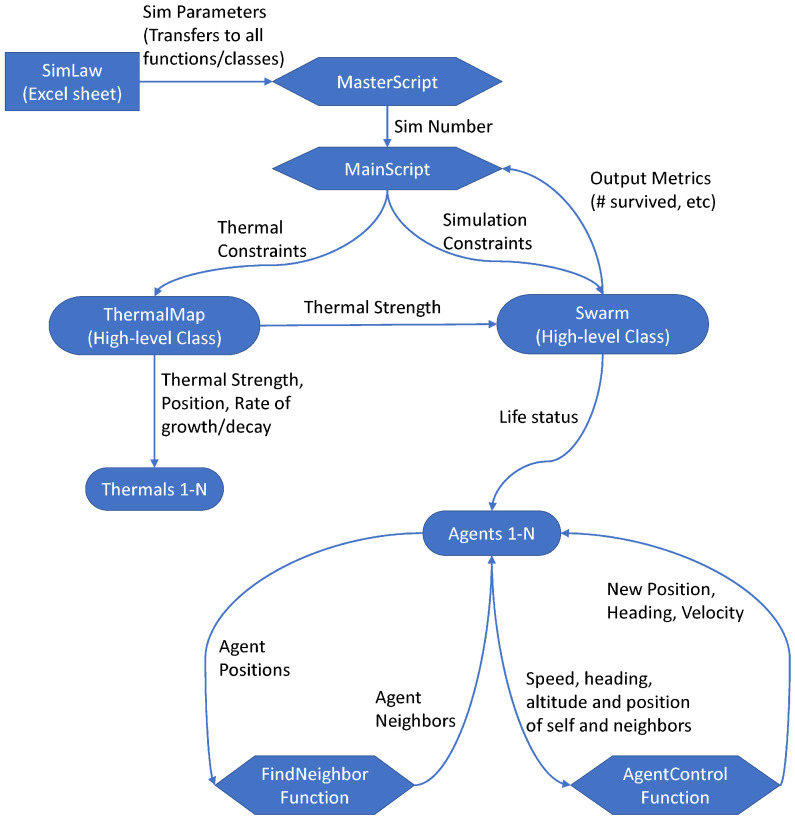
Flowchart of the simulation methodology including high-level scripts, simulation parameters, and intermediary classes.

**Figure 2 biomimetics-08-00124-f002:**
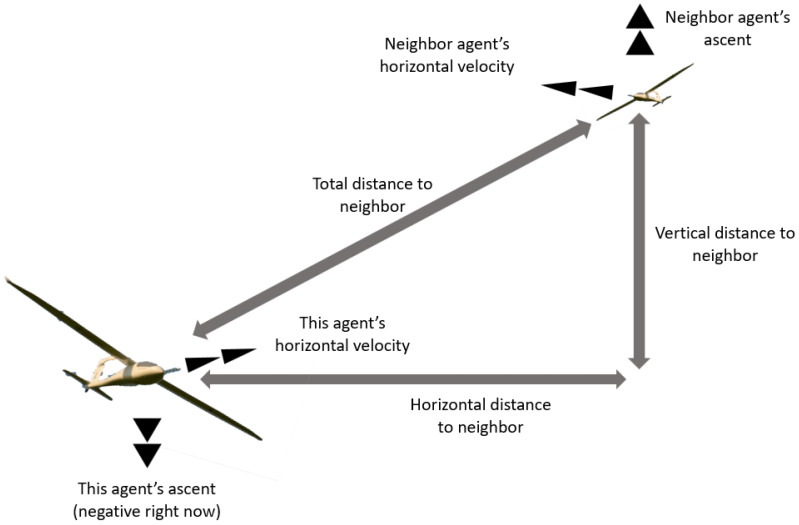
Overview of relevant information between two interacting agents used in calculating agent guidance.

**Figure 3 biomimetics-08-00124-f003:**
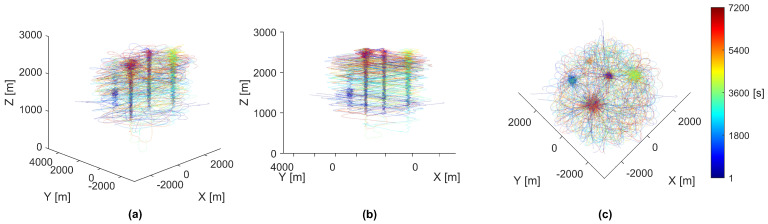
Paths of all 40 agents in a simulation. Each individual agent’s path was colored by time from start to finish measured in seconds, shown by the color-bar on the right side. All panels show different perspectives of the same data. (**a**) The paths rendered from an upper-corner perspective. (**b**) The paths rendered from a side-corner perspective. (**c**) The paths rendered from a top-down perspective.

**Figure 4 biomimetics-08-00124-f004:**
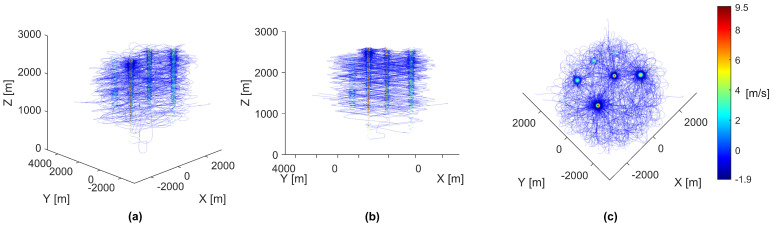
Paths of all 40 agents in a simulation. Each individual agent’s path was colored by vertical speed in meters per second, shown by the color-bar on the right side. All panels show different perspectives of the same data. (**a**) The paths rendered from an upper-corner perspective. (**b**) The paths rendered from a side-corner perspective. (**c**) The paths rendered from a top-down perspective.

**Figure 5 biomimetics-08-00124-f005:**
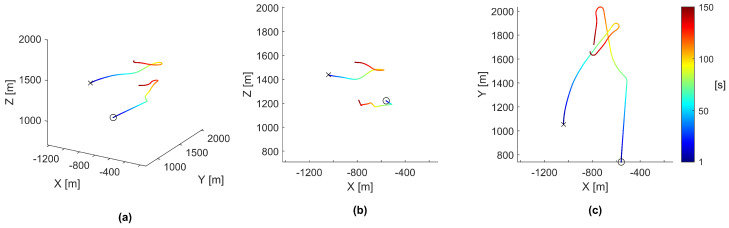
Paths of agents 19 and 30 as they work together to find and exploit a thermal updraft. Agent 19 is marked with an ‘x’ and agent 30 is marked with an ‘o’. Each individual agent’s path was colored by time from start to finish measured in seconds, as shown by the color-bar on the right side. The rendered path data only covers the first 150 s of the simulation. All panels show different perspectives of the same data. (**a**) The paths rendered from an upper-corner perspective. (**b**) The paths rendered from a side perspective. (**c**) The paths rendered from a top-down perspective.

**Figure 6 biomimetics-08-00124-f006:**
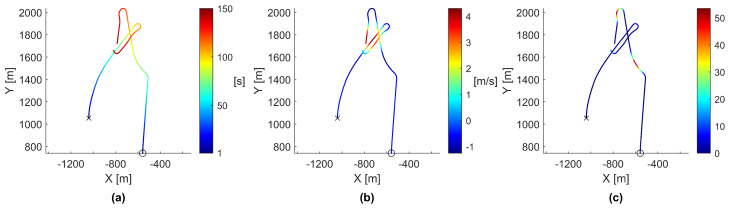
Paths of agents 19 and 30 as they work together to find and exploit a thermal updraft, viewed from a top-down perspective. Agent 19 is marked with an ‘x’, while agent 30 is marked with an ‘o’. The rendered path data only covers the first 150 s of the simulation. (**a**) The paths colored by time in seconds, provided for comparison purposes. (**b**) The paths colored by vertical velocity in meters per second. (**c**) The paths colored by cohesion magnitude to surrounding agents.

**Figure 7 biomimetics-08-00124-f007:**
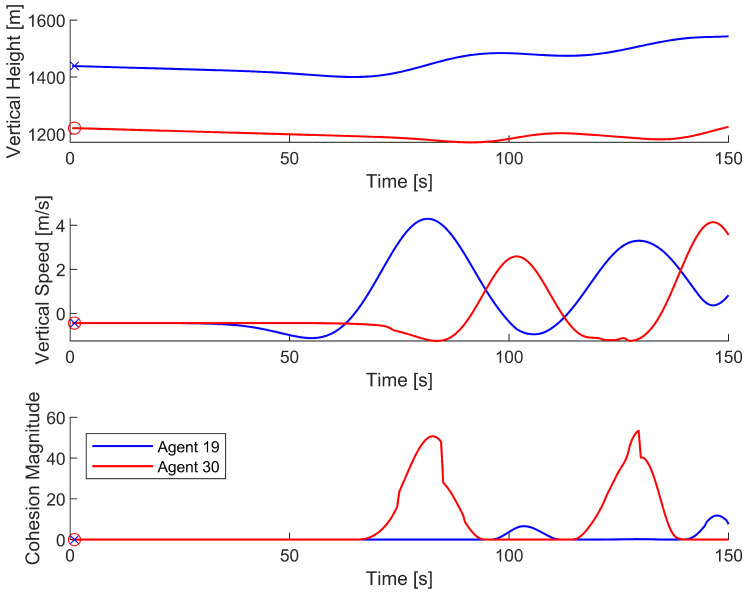
Vertical position, vertical speed, and cohesion magnitude data of agents 19 and 30 as they attempt to find and exploit a thermal updraft. Agent 19 is represented by the blue line and is marked with an ‘x’, while agent 30 is represented by the red line and marked with an ‘o’. The data only covers the first 150 s of the simulation.

**Figure 8 biomimetics-08-00124-f008:**
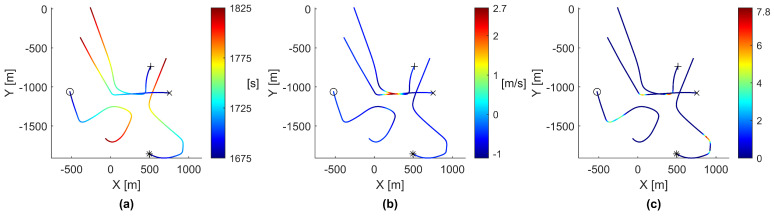
Paths of agents 9, 11, 27, and 29, as they fail to effectively discover a thermal updraft, viewed from a top-down perspective. Agent 9 is marked with an ‘x’, agent 11 with an ‘o’, agent 27 with an ‘*’, and agent 29 with a ‘+’. The rendered path data starts at 1675 s and continues until 1825 s. (**a**) The paths colored by time in seconds. (**b**) The paths colored by vertical velocity in meters per second. (**c**) The paths colored by cohesion magnitude to surrounding agents.

**Figure 9 biomimetics-08-00124-f009:**
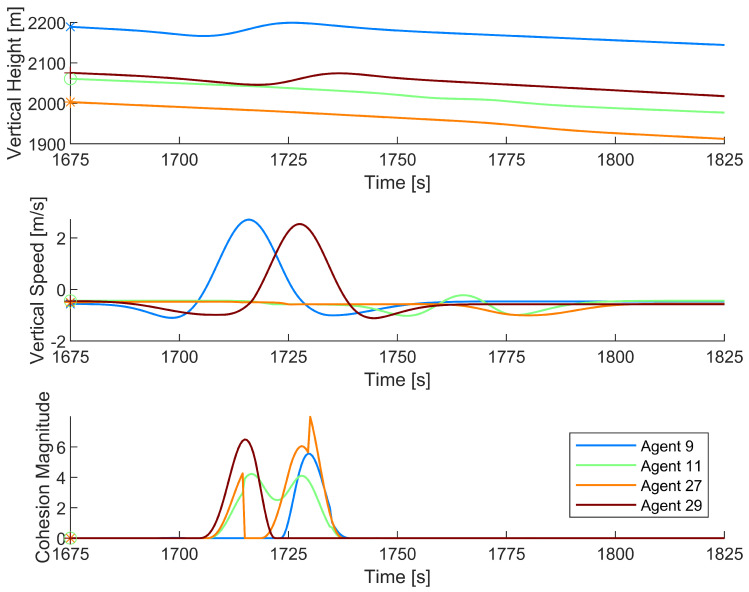
Vertical position, vertical speed, and cohesion magnitude data of agents 9, 11, 27, and 29, as they fail to effectively discover a thermal updraft. Agent 9 is marked with an ‘x’, agent 11 with an ‘o’, agent 27 with an ‘*’, and agent 29 with a ‘+’. The rendered path data starts at 1675 s and continues until 1825 s.

**Figure 10 biomimetics-08-00124-f010:**
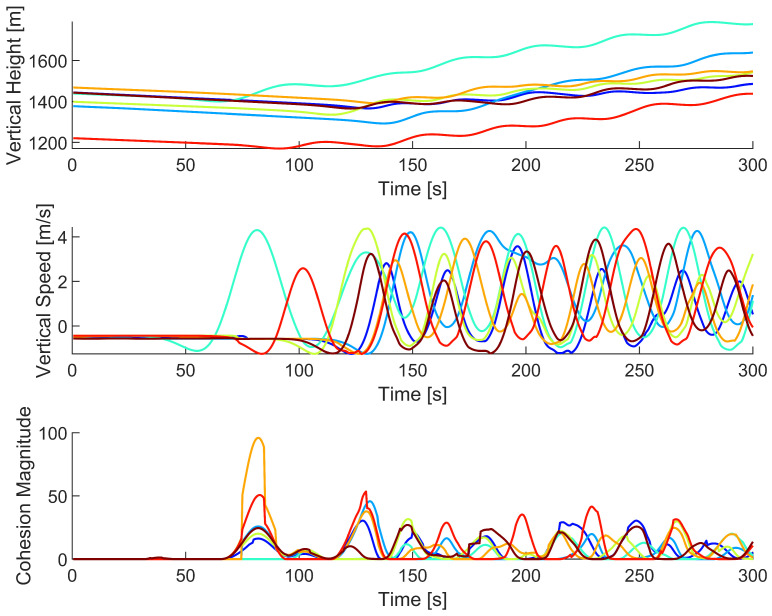
Vertical position, vertical speed, and cohesion magnitude data of a group of agents as they discover and exploit a thermal updraft. The included agents are agents 7, 13, 19, 21, 28, 30, and 36, chosen due to their close proximity and exploitation of the chosen thermal updraft. The data only covers the first 300 s of the simulation.

**Figure 11 biomimetics-08-00124-f011:**
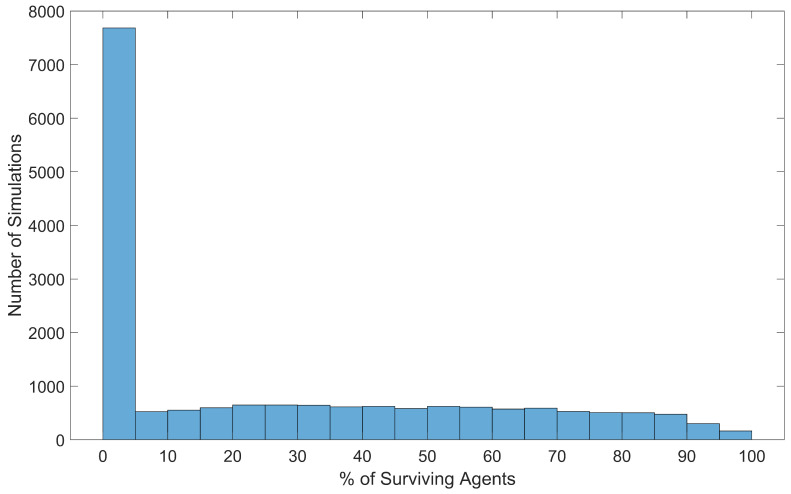
Distribution of swarm survival at the end of each simulation, across all 18,000 simulations.

**Figure 12 biomimetics-08-00124-f012:**
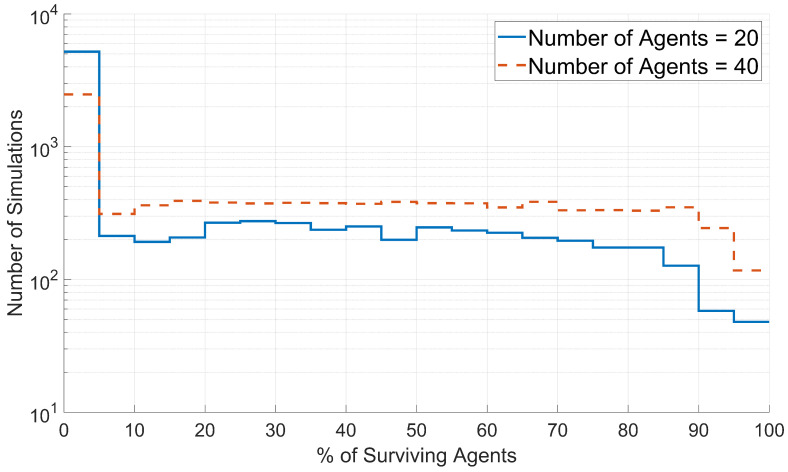
Distribution of swarm survival for simulations with 20 agents and 40 agents. The number of simulations for a given percentage of surviving agents is plotted on a logarithmic scale.

**Figure 13 biomimetics-08-00124-f013:**
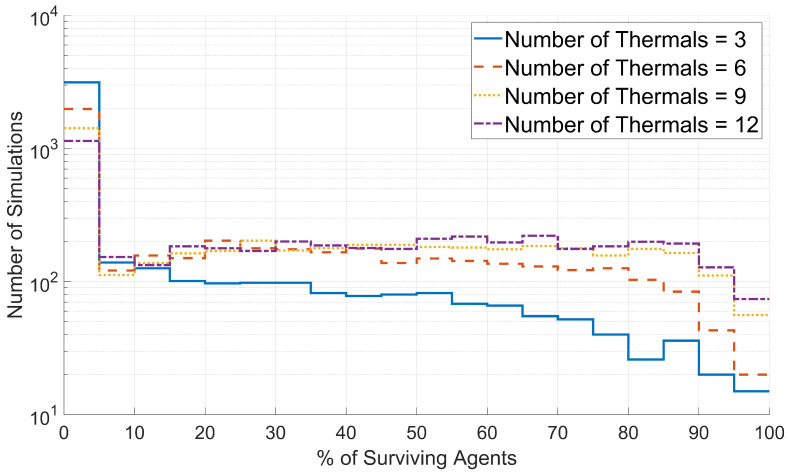
Distributions of swarm survival for simulations with 3, 6, 9, and 12 thermal updrafts. The number of simulations for a given percentage of surviving agents is plotted on a logarithmic scale.

**Figure 14 biomimetics-08-00124-f014:**
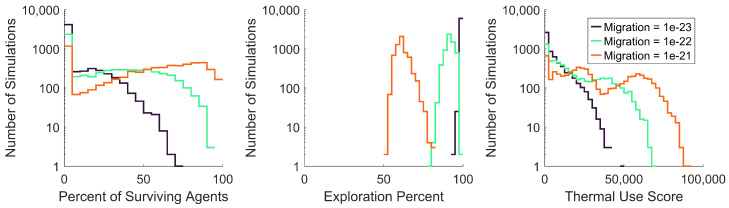
Distribution of swarm survival, exploration, and thermal use for all values of migration, including $1\times 10^{-23}$, $1\times 10^{-22}$, and $1\times 10^{-21}$. All histogram values showing a number of simulations are plotted on a logarithmic scale.

**Figure 15 biomimetics-08-00124-f015:**
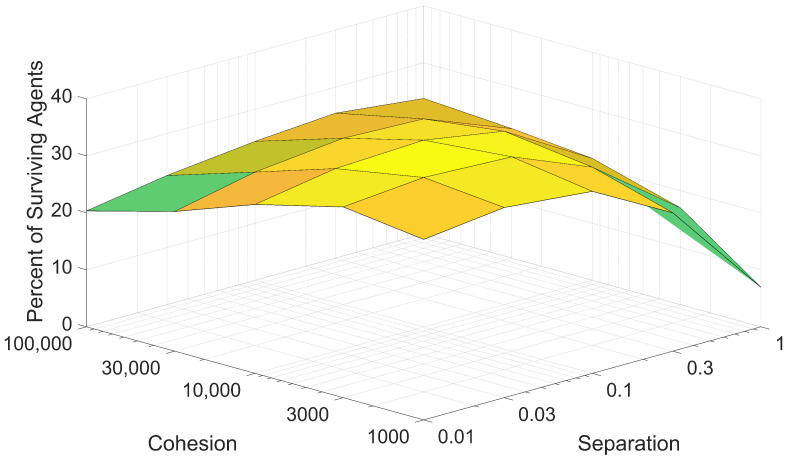
Mean swarm survival plotted against cohesion and separation values. Cohesion and separation values are plotted on logarithmic scales.

**Figure 16 biomimetics-08-00124-f016:**
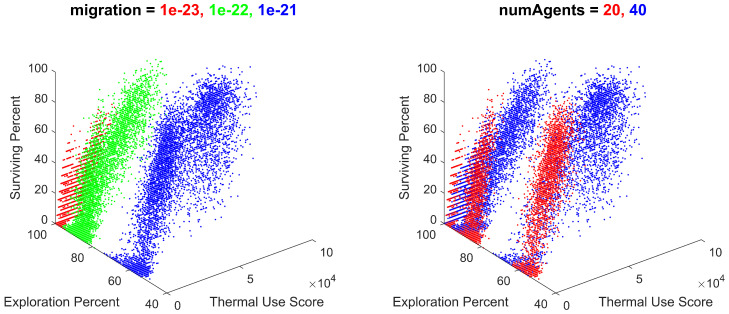
All batch simulation data plotted for thermal use, exploration percentage, and survival percentage, colored for migration values of $1\times 10^{-23}$, $1\times 10^{-22}$, and $1\times 10^{-21}$ and for initial number of agents values of 20 and 40. Note the distinct clumping within the plots, resulting from each unique value of migration and initial number of agents.

**Figure 17 biomimetics-08-00124-f017:**
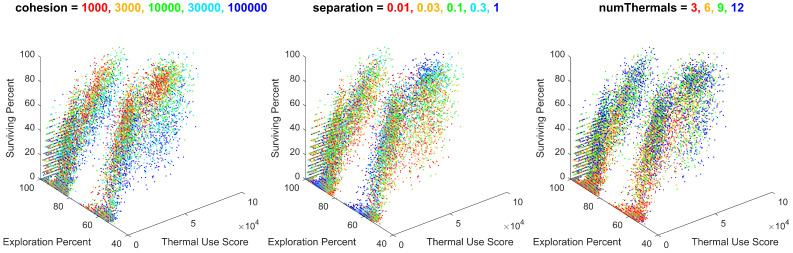
All batch simulation data plotted for thermal use, exploration percentage, and survival percentage, colored for cohesion values of 1000, 3000, 10,000, 30,000, and 100,000; separation values of 0.01, 0.03, 0.1, 0.3, and 1; and number of thermals of 3, 6, 9, and 12. Note the color gradients within the plots, resulting from each unique value of cohesion, separation, and number of thermals.

**Figure 18 biomimetics-08-00124-f018:**
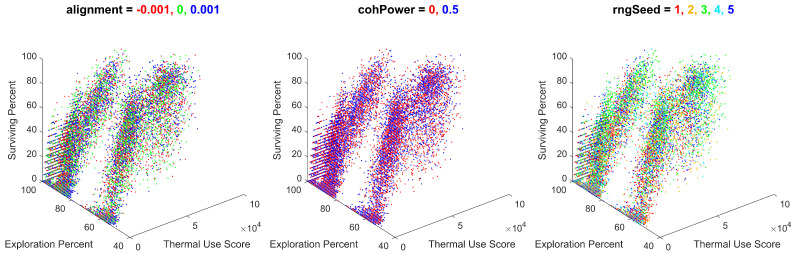
All batch simulation data plotted for thermal use, exploration percentage, and survival percentage, colored for alignment values of −0.001 and 0, 0.001, cohesion power values of 0 and 0.5, and random number generator seeds of 1, 2, 3, 4, or 5. Note the lack of any identifiable trend between the results and each unique value for the alignment, cohesion power, and RNG seed.

**Table 1 biomimetics-08-00124-t001:** Overview of parameters varied in latest simulation batch, including a description, the values tested, the number of values of each parameter, and the number of resulting combinations of all parameters.

Parameter	Description	Tested Values	Number of Values
numAgents	Number of agents at simulation start	20, 40	2
numThermals	Number of thermal updrafts	3, 6, 9, 12	4
rngSeed	RNG seeds for repeatability	1, 2, 3, 4, 5	5
cohesion	Cohesion gain	$1\times 10^{3}$, $3\times 10^{3}$, $1\times 10^{4}$, $3\times 10^{4}$, $1\times 10^{5}$	5
separation	Separation gain	$1\times 10^{-2}$, $3\times 10^{-2}$, $1\times 10^{-1}$, $3\times 10^{-1}$, 1	5
alignment	Alignment gain	$-1\times 10^{-3}$, 0, $1\times 10^{-3}$	3
migration	Migration gain	$1\times 10^{-23}$, $1\times 10^{-22}$, $1\times 10^{-21}$	3
cohPower	Exponent of distance for cohesion	0, 0.5	2
Total Combinations			18,000

**Table 2 biomimetics-08-00124-t002:** An example of the recorded data for each individual simulation, including the varied input parameters and the resultant output metrics.

Input/Output	Recorded Variable Name	Recorded Value
Input	Cohesion	1000
Input	Separation	0.3
Input	Alignment	0.001
Input	Cohesion Power	0
Input	Migration	$1\times 10^{-21}$
Input	Number of Thermals	9
Input	Number of Agents	40
Input	RNG Seed	2
Output	Number of Surviving Agents	40
Output	Height Score	$5.5318\times 10^{+8}$
Output	Thermal Use Score	56708
Output	Exploration Percentage	67
Output	Flight Time	288,000
Output	Collision Deaths	0
Output	Ground Deaths	0
Output	Final Height Maximum	2271.22
Output	Final Height Minimum	752.903
Output	Final Height Average	2001.57

## Data Availability

The GitHub repository in which all work supporting the results of this paper can be found here: https://github.com/wvu-robotics/REU_MatlabSim/tree/main/matlab/REU_2022/Topic_3_Soaring.

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
