# Peer review of "Analysis of UAV Thermal Soaring via Hawk-Inspired Swarm Interaction"

_biomimetics, 2023, doi:10.3390/biomimetics8010124_

Round 1

Reviewer 1 Report

The paper observes the interaction behavior of the natural eagle swarm, analyzes its movement rules and contributes to providing insights as to which behavioral rules for a swarm of UAVs result in better collective usage of thermals. Finally, experimental analysis is carried out by simulation experiments. However, there are still some deficiencies in this paper:

1、 In the introduction section, the research on related methods is not comprehensive. Suggest further research on relevant papers.

2、 The "energy management" in the title has not been deeply reflected and analyzed in the paper. We suggest adding relevant content or modifying relevant wording.

3、 We suggest improving the description of the conclusion, adding experimental statistics, and further indicating the significance of the study.

Reviewer 2 Report

In general, the paper is well-written, and the idea is interesting. My main concerns are:

1 - The paper must present more motivations in the introduction. 

2 - Besides, a comparison with the state-of-the-art and more related works must be performed;

3 - Insert a flowchart at the beginning of the methodology to give the reader a summarized notion of what is being proposed;

4 - Described the hardware used in the simulations and how the tests were performed;

5 - More results are required. Please, provide it.

6 - Give a discussion regarding the results compared to the literature.

7 - Figures 12 and 13 must be of better quality. Please, revise the figures of the manuscript to let the text of the figures in the same size and font of the text.

Reviewer 3 Report

1. The use of a flat surface for modeling caused certain doubts. In reality, the formation of thermals is particularly strongly influenced by the unevenness of the earth's surface, the nature of the land cover, surface sediments, and hydrographic elements. A simplified version - a flat homogeneous surface - makes solving the problem easier, but the reality can have a significant impact on the actions of a fixed-wing UAV swarm.

2. Correctly selected parameters that simulate the behavior of a flock of birds: cohesion, separation, and alignment. I believe that such a complex of factors ensures the necessary autonomy and communality of the UAV swarm.

3. The authors of the article presented convincing data reflecting the simulation paths of fixed-wing UAV flights (Figures 2, 3). It also shows in an argumentative manner how individual UAVs can and do respond to "leaders", i.e. those who detect and use the lifting power of thermal thermals (Fig. 6).

4.

It is to be congratulated that the manuscript authors in the Conclusions section (which is also the Discussions), very clearly identified the factors necessary for more detailed analysis in the future, which were more approximated in the presented research. They correctly observed that the movement of the UAV swarm is influenced by the relatively fast natural dynamics of thermals in time and space. They correctly note that simulations performed using a wider range of parameters would help to more accurately study the behavior of UAV swarms under real-world conditions. I wish the authors to delve into the additional parameters describing the shields of the UAV swarm in the future and include them in the model.

Round 2

Reviewer 2 Report

The paper has been improved. However, the authors must:

- Improve the quality of Figures 2, 12, 13, and 16

- In Figure 3, put the label (b) in the same line as (a) and (c). 

- Equations with "*" are convolution? I believe that is not. Change for multiplication characters. Besides, vectors and matrices are in bold. Fix it.

- Discuss the obtained results with other works of literature.

- English must be checked to improve readability. 
